# A novel method of screening combinations of angiostatics identifies bevacizumab and temsirolimus as synergistic inhibitors of glioma-induced angiogenesis

**Michael I. Dorrell**[1]*, **Heidi R. Kast-Woelbern**[1], **Ryan T. Botts**[2], **Stephen A. Bravo**[3], **Jacob R. Tremblay**[3], **Sarah Giles**[3], **Jessica F. Wada**[3], **MaryAnn Alexander**[3], **Eric Garcia**[3,4], **Gabriel Villegas**[3], **Caylor B. Booth**[5], **Kaitlyn J. Purington**[5], **Haylie M. Everett**[5], **Erik N. Siles**[5], **Michael Wheelock**[5], **Jordan A. Silva**[3,4], **Bridget M. Fortin**[3,4], **Connor A. Lowey**[3,4], **Allison L. Hale**[3,4], **Troy L. Kurz**[3], **Jack C. Rusing**[3], **Dawn M. Goral**[3], **Paul Thompson**[3], **Alec M. Johnson**[3], **Daniel J. Elson**[3], **Roujih Tadros**[3,4], **Charisa E. Gillette**[3,4], **Carley Coopwood**[3,4], **Amy L. Rausch**[3,4], **Jeffrey M. Snowbarger**[3,4]

**1** Department of Biology, Point Loma Nazarene University, San Diego, CA, United States of America,
**2** Department of Mathematical, Information, and Computer Sciences, Point Loma Nazarene University, San Diego, CA, United States of America, **3** Department of Biology, Dr. Michael Dorrell's Lab, Point Loma Nazarene University, San Diego, CA, United States of America, **4** Department of Biology, Dr. Heidi R. Kast-Woelbern's Lab, Point Loma Nazarene University, San Diego, CA, United States of America, **5** Department of Mathematical, Information, and Computer Sciences, Dr. Ryan Bott's Lab, Point Loma Nazarene University, San Diego, CA, United States of America

* mdorrell@pointloma.edu

**Data Availability Statement:** The data for the CAM experiments, including all the supplemental data

## Abstract

Tumor angiogenesis is critical for the growth and progression of cancer. As such, angiostasis is a treatment modality for cancer with potential utility for multiple types of cancer and fewer side effects. However, clinical success of angiostatic monotherapies has been moderate, at best, causing angiostatic treatments to lose their early luster. Previous studies demonstrated compensatory mechanisms that drive tumor vascularization despite the use of angiostatic monotherapies, as well as the potential for combination angiostatic therapies to overcome these compensatory mechanisms. We screened clinically approved angiostatics to identify specific combinations that confer potent inhibition of tumor-induced angiogenesis. We used a novel modification of the *ex ovo* chick chorioallantoic membrane (CAM) model that combined confocal and automated analyses to quantify tumor angiogenesis induced by glioblastoma tumor onplants. This model is advantageous due to its low cost and moderate throughput capabilities, while maintaining complex *in vivo* cellular interactions that are difficult to replicate *in vitro*. After screening multiple combinations, we determined that glioblastoma-induced angiogenesis was significantly reduced using a combination of bevacizumab (Avastin®) and temsirolimus (Torisel®) at doses below those where neither monotherapy demonstrated activity. These preliminary results were verified extensively, with this combination therapy effective even at concentrations further reduced 10-fold with a CI value of 2.42E-5, demonstrating high levels of synergy. Thus, combining bevacizumab and temsirolimus has great potential to

for drugs that did not show activity in our model and combinations that were not synergistic, as well as the CI calculations for the Avastin and Torisel combinations, can be found shared from OSF storage, project "Combination Angiostatic Therapy_PLoS One-2021" at: https://osf.io/cd46p/ The computational files required for the R program to run the automated quantification can be found at: https://github.com/rbotts/TumorQuant.

**Funding:** Dorrell MI: Point Loma Nazarene University - Annual in-house Alumni Grants and Research and Special Projects grants. Annual funding from Alumni Associates. Botts RT: NIH AREA grant (NIGMS) #1R15GM102995-01A1. Point Loma Nazarene University - In-house Research and Special Projects grants. Annual funding from Alumni Associates. Kast-Woelbern HR: Point Loma Nazarene University - Annual in-house Alumni Grants and Research and Special Projects grants. Annual funding from Alumni Associates. All other authors were undergraduates in MD, RB, or HK's labs. The funders had no role in study design, data collection and analysis, decision to publish, or preparation of the manuscript. RASP grant numbers: Dorrell alumni = GR-19690 Dorrell RASP 2018 – 2019 = GR-20960 Dorrell RASP 2020 - 2021 = GR-22190 Heidi alumni (Angiogenesis) = GR-20470 Heidi RASP = GR-22150.

**Competing interests:** The authors have declared that no competing interests exist.

**Abbreviations:** CAM, Chorioallantoic membrane; GBM, glioblastoma multiforme; PD-1, Programmed Cell Death Protein 1; CTLA4, Cytotoxic T-Lymphocyte Associated Protein 4; NV, neovascularization; VEGF, Vascular Endothelial Growth Factor; mTOR, Mammalian target of rapamycin; CI, Combination Index.

increase the efficacy of angiostatic therapy and lower required dosing for improved clinical success and reduced side effects in glioblastoma patients.

## Introduction

Cancer is one of the leading causes of death, affecting almost 40% of individuals in their lifetime [1]. Glioblastoma multiforme (GBM) is a deadly brain cancer due to its highly aggressive and invasive nature and is one of the most difficult human malignancies to treat [2, 3]. Despite extensive research and progress in neuro oncology and diagnostics, there remains a great need for improved treatments that potently reduce tumor growth and minimize side effects. The use of the chemotherapeutic temozolomide in conjunction with radiation treatment following surgical resection has elongated median life-expectancies for patients with GBM, but survival rates are still typically between one and two years following diagnosis [4, 5]. More recent research has demonstrated that immunotherapy options, such as Programmed Cell Death Protein-1 (PD-1) or Cytotoxic T-Lymphocyte Associated Protein 4 (CTLA-4) inhibition, may prove useful for treating brain tumors [6, 7], but these have not yet succeeded in phase 3 clinical trials [8]. Thus, further research is required to continue informing treatment options for patients diagnosed with these devastating tumors.

Neovascularization (NV) is a hallmark of most, if not all, cancers including glioblastoma [9], and contributes to both tumor growth and metastasis [10, 11]. Tumors initiate NV through the "angiogenic switch", in which previously quiescent vessels are prompted to proliferate, bringing oxygen and nutrients required for continued tumor growth. This is a key aspect of tumor progression and metastasis, and it has been extensively demonstrated that blocking tumor NV limits tumor growth [10, 12–14]. While angiogenesis is critical during development, its role in adulthood is limited with the main exceptions being during major wound healing, endometrial changes due to the menstrual cycle [15], and pregnancy [16]. Adult angiogenesis is also indicated in particular diseases such as cancer [17], retinal eye disease [18] and atherosclerosis [19]. As such, angiostatic treatments are largely disease-specific in otherwise healthy adults, lacking the major side effects associated with chemotherapy, which generally does not differentiate between tumor and healthy rapidly dividing cells. Many angiostatic molecules have been described [12, 14, 20, 21], and are valuable adjuncts to conventional chemotherapy, reducing tumor loads and prolonging survival. Unfortunately, cancer treatments using angiostatics as stand-alone monotherapies have yet to fulfill their clinical potential [22–25]. Because of this, research has begun to focus on the development of other treatments, such as immunotherapies [26]. However, if successful angiostatic treatments can be found, these treatment options would still offer several advantages including reduced side effects and broader intervention across cancer types by targeting the vasculature rather than the heterogeneous cancers. Even if stand-alone angiostatic treatment is not viable, potent angiostatic regimens can offer viable adjunct therapy to other treatment modalities, potentially reducing the doses and side effects.

Angiogenesis is a fundamental biological process essential to survival of the organism. As such, redundant mechanisms have evolved to facilitate new blood vessel growth and angiogenesis is initiated by the overlapping activation of multiple pathways *in vivo*. These compensatory mechanisms limit the therapeutic potential of anti-angiogenic monotherapies [14, 27, 28]. We have previously demonstrated this using models of developmental and pathological angiogenesis in the retina, and in glioblastoma brain tumors. In these models, the use of angiostatic monotherapies resulted in upregulation of compensatory stimuli, thus limiting the

effectiveness of any single drug [28]. This compensation, and ultimately angiogenesis, was blocked by the combined use of multiple angiostatics, each targeting distinct aspects of the angiogenic process. This combination therapy prevented compensatory upregulation and potently inhibited angiogenesis in both ocular and tumor models, even at low doses, resulting in nearly complete inhibition of NV and significantly increasing life expectancy in an aggressive model of glioblastoma [28]. However, not all combinations have synergistic effects. In the early proof-of-principle study, most combinations lacked synergy, particularly those using different mechanisms to target vascular endothelial growth factor (VEGF) initiation of NV. Unfortunately, the three factors used successfully in combination in that proof-of-principle study, the VEGF antagonist [29], integrin inhibitor [30], and a small fragment of tryptophan tRNA synthesis (mechanism unknown [31]) are not clinically-approved therapies, although variations of the VEGF antagonist are currently approved. Many studies since then have focused on combination therapy to overcome the ineffective monotherapies, particularly as tumors become increasingly resistant to VEGF inhibition [32]. Further work is needed to identify various combinations of approved angiostatics that have synergistic effects and can overcome these compensatory mechanisms to prevent glioblastoma-induced angiogenesis.

In this study, we screened multiple combinations of approved angiostatics to identify synergistic combinations that effectively block glioma-induced tumor vascularization similar to the effects observed in our earlier study [28]. We developed and utilized a novel adaptation to the commonly used *ex ovo* chick chorioallantoic membrane model (CAM) [33, 34] whereby tumor vascularization can be imaged and auto-quantified to objectively compare tumor angiogenesis levels from different treatment groups [35]. Through this screening approach, we determined that bevacizumab (Avastin®) and temsirolimus (Torisel®) work synergistically to inhibit tumor-induced NV at doses that are completely ineffective individually. Both drugs are currently available and approved for use in the treatment of certain cancers and thus could potentially be applied in combination to improve the outcome for patients with glioblastoma.

## Materials and methods

### *Ex ovo* CAM preparation

Preparation of the *ex ovo* CAMs followed published protocols [33, 35]. Briefly, fertilized eggs were obtained and incubated for three days at 37˚C in humidified chambers. On day 3 post-fertilization (3dpf), the embryonated egg contents were removed from the shell as previously described, and grown *ex ovo* for 1 week until 10dpf when the tumor onplants were applied. Each of six tumor onplants was gently placed at evenly spaced intervals onto the chorioallantoic membrane approximately one-half inch away from the central embryo (Fig 1). *Ex ovo* embryos were incubated for a further 3 days to allow tumor angiogenesis to occur, at which point the embryos were perfused, euthanized and the tumor onplants were excised by cutting out each onplant with the underlying CAM. Excised onplants were prepared, imaged, and tumor vasculature was quantified (see Fig 1 and information below for details on each step).

**Cell lines.** Rat 9L or human U87 glioblastoma cells were used to create the tumor onplants. Rat 9L cells were obtained from Dr. Martin Friedlander and Dr. Faith Barnett at the Scripps Research Institute. These cells were a frozen aliquot from those used in previous publications [28, 36]. Human U87 glioblastoma cells were obtained from ATCC (ATCC HTB-14), along with human cell-line authentication. Low-passage aliquots were frozen in liquid nitrogen. Cells were grown and maintained in high glucose DMEM media with L-glutamine (ThermoFisher cat# 11965092) and passaged according to standard protocol, ensuring that confluency never exceeded 80% and cells never exceeded passage ten to prevent unwanted cellular changes. Twenty-four hours prior to use in the study, cell media was replaced with

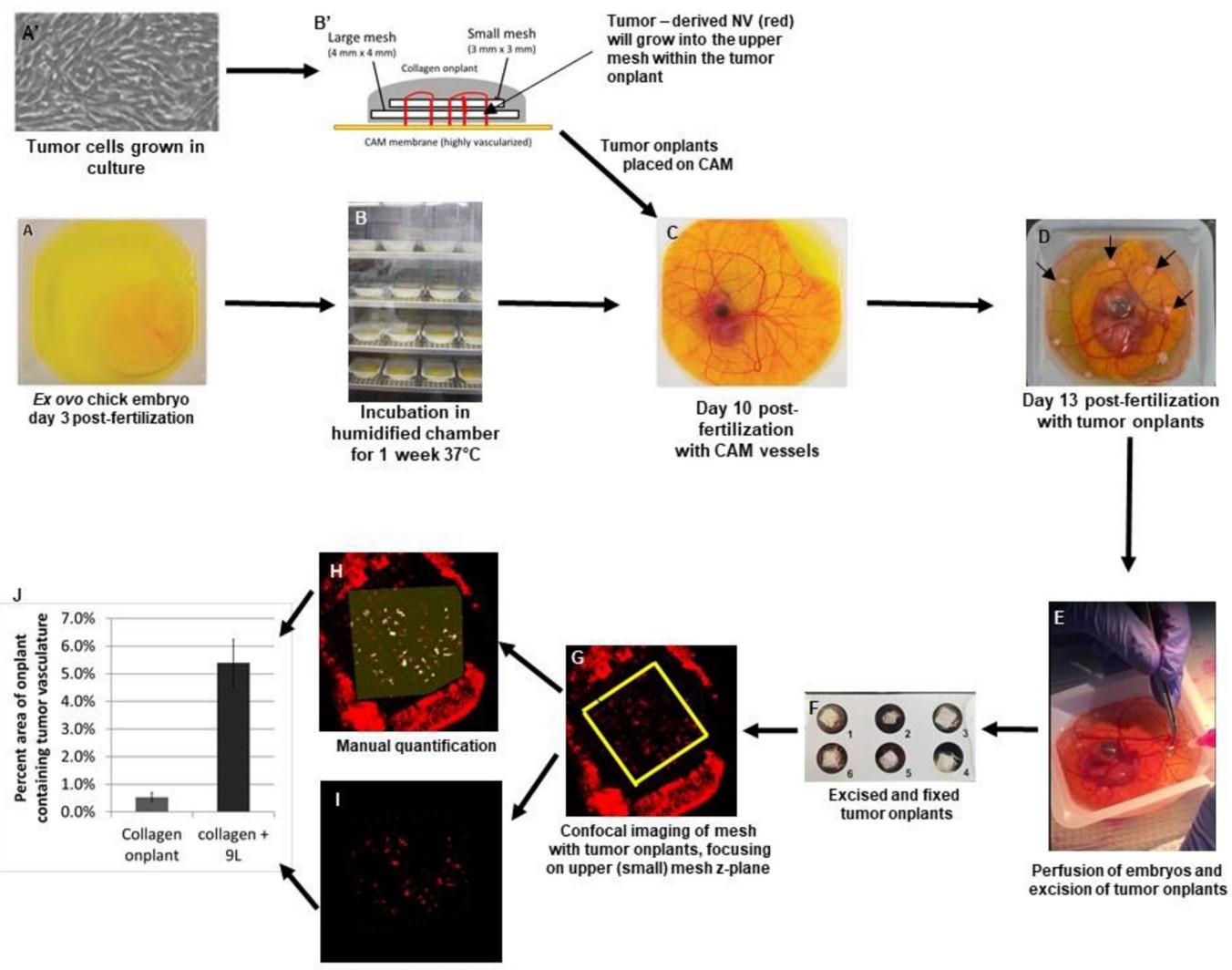

**Fig 1. Description of the *Ex ovo* CAM model.** (A) Fertilized eggs day 3 post-fertilization (3dpf) were removed from the shell and grown *ex ovo*. (B) Embryos were grown for an additional 7 days at 37°C in humidified chambers. (A') Meanwhile, tumor cells were grown in culture and then (B') incorporated along with various drugs into 3-dimensional collagen 1 based onplants incorporating a larger and smaller mesh whereby tumor-derived NV (red lines) will sprout from the CAM vessels and branch upward into the onplant. (C) On day 10 post-fertilization, six tumor onplants were applied onto each CAM at evenly spaced intervals, using at least 5 different CAMs per group (≥30 replicates total). (D) Image of a day 13 post-fertilization *ex ovo* embryo with six tumor onplants (indicated by arrows). (E) On day 13, CAM vessels were perfused with DIL, embryos were euthanized and tumor onplants were excised. (F) Excised onplants were fixed to a 6-well microscope slide and imaged using confocal microscopy (G). The area of NV within the upper mesh was then quantified manually (H) or by automated quantification (I). (J) The onplants containing tumor cells resulted in 5 – 10x the amount of quantified vasculature within the onplant compared to the negative control onplants.

serum-free, low glucose DMEM media (ThermoFisher cat# 11885076) to activate the tumor cells towards a more energy depleted state, thus optimizing pro-angiogenic activity.

**Preparation of tumor onplants.** After 24 hours of growth in serum-free, low-glucose media, 9L or U87 tumor cells at ~70% confluency (optimal growth phase) were removed using enzyme-free dissociation media (Gibco). Cells and angiostatics were incorporated into onplants at a cell density of 1000 cells/μL within a final concentration of 2.1 mg/mL type I collagen (BD Biosciences). 30μL collagen droplets were polymerized as previously described [37] immersing two separate small meshes (180-μm openings) stacked on top of each other, with a

larger (4 mm x 4 mm) square of mesh beneath a smaller (3 mm x 3 mm) square. These 3D tumor onplants were grafted onto the chorioallantoic membrane (CAM). Each experiment included a negative control group with collagen onplants containing no tumor cells or drug (vehicle—phosphate buffered saline (PBS) controls) and a positive control with collagen onplants containing tumor cells alone (no drug). Each collagen onplant and its contents (PBS, tumor, tumor + drug, etc.) encompassed the entirety of the two meshes. The use of these two meshes facilitated focusing on the top mesh during confocal microscopy imaging to distinguish tumor vasculature from the naturally developing CAM vessels underneath; tumor vasculature must grow upward from the underlying horizontal CAM vessels through the bottom mesh into the onplant and the z-plane of the top mesh. Tumor onplants were placed upon the CAMs of day 10 post-fertilization embryos and tumor angiogenesis was analyzed three days later. These methods have been extensively used in publication for the study of tumor angiogenesis and the effects of angiostatics [34]. The novel modification of published protocols comes during quantification of the tumor angiogenesis (details below).

**Preparations of angiostatics for screening.** See Table 1 for details on the angiostatic drugs tested. Angiostatic drugs were incorporated directly into the collagen onplants along with the tumor cells. For each individual drug, a broad range of doses were tested until a dose that significantly affected tumor NV in the CAM model was found (Fig 3), thus determining which drugs demonstrated efficacy in the CAM model with rat 9L tumor induction (Table 1). After initially obtaining broad dosing information, the dosing range was refined to determine the maximum non-effective dose for each angiostatic in the CAM model (Fig 3). These doses were confirmed in at least two additional experiments. Within experiments, each group comprised a minimum of five different chick embryos with six onplants each (at least 30 total replicates) to account for technical variation between onplants and natural biological variation.

Thalidomide (Sigma Aldrich) was dissolved at 100x concentration in ethanol followed by further dilution in PBS (final concentration of ethanol was < 0.01%; vehicle (PBS) control was

**Table 1. List of angiostatics obtained and tested in our CAM model for screening for synergistic combinations.**

| Name: | Obtained from: | Brief description of activity | Indication(s) | Delivery | FDA status |
|---|---|---|---|---|---|
| *Bevacizumab (Avastin®)* | Genentech (clinical grade) | Monoclonal antibody that blocks VEGF-A and prevents receptor binding | colorectal, small cell lung, glioblastoma, kidney | IV | approved |
| *Temsirolimus (Torisel®)* | Wyeth (clinical grade) or Tocris biosciences | Blocks mTOR, thus blocking both angiogenesis (mTOR activates HIF-1α) and cell cycle (regulates cyclins) | Advanced renal cell carcinoma | IV | approved |
| *Thalidomide* | Sigma-Aldrich | Blocks angiogenesis and may be linked to TNF-α inhibition. | Multiple myeloma | Oral tablet | approved |
| **Pazopanib (Votrient®)** | Novartis (clinical grade) | Multi-targeted tyrosine kinase inhibitor; inhibits VEGFR-1, VEGFR-2, VEGFR-3, PDGFR, and c-kit | Renal cell carcinoma | Oral tablet | approved |
| **Sunitinib (Sutent®)** | Pfizer (clinical grade) | Small-molecule inhibitor of multiple receptor tyrosine kinases | Renal cell, gastro-intestinal stromal | Oral capsule | approved |
| Cetuximab (Erbitux®) | Lilly (clinical grade) | Mouse/human monoclonal antibody chimera inhibits Epidermal growth factor receptor activity | Colorectal, head and neck | IV | approved |
| Sorafenib (Nexavar®) | Sigma-Aldrich (Y0002098) | Small molecule inhibitor of multiple tyrosine protein kinases. Unique in targeting the MAP kinase pathway | Advanced renal cell carcinoma | Oral tablet | approved |
| Erlotinib hydrochloride (Tarceva®) | Sigma-Aldrich (CDS022564) | Inhibits epidermal growth factor receptor (receptor tyrosine kinase antagonist) | Non-small cell lung, pancreatic | Oral tablet | approved |
| RGB peptide ~Cilengitide | Sigma-Aldrich (A8052) | Blocks $\alpha_v\beta_3$ and $\alpha_v\beta_5$ activity affecting endothelial cell migration and survival | Glioblastoma | IV | Phase III trials |

Drugs in bold type indicate the drugs that demonstrated significant angiostatic activity in our model system and were therefore included in the screen for combination therapy.

adjusted accordingly). Bevacizumab (Avastin®), temsirolimus (Torisel®), pazopanib (Votrient®), sunitinib (Sutent®), and cetuximab (Erbitux®) were prepared according to the clinical instructions provided by the manufacturer and diluted in PBS to the final concentrations. In each case, the initial drug's solution was diluted by 100-fold (minimum) in PBS. After early experiments, temsirolimus powder (>99% pure) was used in place of Torisel® for cost purposes, but activity in our assay was found to be equivalent. Sorafenib (Nexavar®), and erlotinib hydrochloride (Tarceva®) were purchased directly from Sigma-Aldrich and formulations of these compounds, as well as temsirolimus powder were initially solubilized in 100% dimethyl sulfoxide (DMSO), followed by extensive dilution in PBS. Final doses included less than 0.1% DMSO. All PBS (vehicle) controls were adjusted accordingly for those experiments to account for any differences in solution preparation. The cyclic RGD peptide, cilengitide, was not commercially available so we used standard RGD peptides (Sigma Aldrich, A8052), which were solubilized and diluted in PBS immediately prior to use. All drugs and drug combinations were incorporated into the cell / collagen onplant preparations during polymerization, prior to addition to the CAM embryos.

## Analysis of tumor angiogenesis

**Vascular perfusion.** After the tumor onplants had grown on the CAM for three days (10dpf– 13dpf), the chick embryos were intravenously perfused with $DiIC_{18}(3)$, allowing for intercalation into the endothelial membranes and subsequent fluorescence of the vasculature. The DiI staining solution was prepared as described [38]. Briefly, a 6.4 mM stock solution of DiI was made by dissolving 10 mg of DiI crystal in 1.67 mL of 100% ethanol. Immediately before use, a 130 μM working solution was made by adding 200 μL of DiI stock solution to 10 mL of 5% w/v glucose in PBS. Approximately 1.5 mL of the DiI solution was perfused into the chick until bleeding was observed due to excess volume. After allowing the DiI to continue flowing through the vasculature for two additional minutes, the chick embryo was euthanized on ice. After excision, the onplants were briefly washed in PBS followed by fixation in 4% paraformaldehyde for 30 minutes. Following a second brief wash in PBS the onplants were mounted onto six-well slides with the small mesh oriented upward and a small amount of anti-fade reagent (Invitrogen). Onplants from the different groups were placed on slides based on a random number generator such that all further imaging and quantification was performed by individuals "masked" as to the identity of the treatment group for each onplant imaged and quantified.

**Confocal imaging.** Confocal microscopy was used to image the tumor vasculature by focusing on the z-plane of the smaller, 3 x 3 mm upper mesh. The entire onplant was imaged by 4x4 tiling at 50X total magnification using 500 pixels/inch using a Zeiss LSM 210. The images were saved as tiff files and the area of the upper mesh was outlined in yellow to allow for subsequent quantification of tumor vasculature within the onplant; we used the yellow polygon drawing tool in the Zeiss confocal software (Fig 1). All imaging and subsequent quantification was performed by trained, masked individuals. Note that to use our available automated quantification, the fluorescent vasculature must be imaged in red, and the upper mesh outlined in yellow (see below).

**Manual vascular quantification.** For manual vascular quantification, Photoshop was used. The area of the upper mesh was selected and colored with 80% transparency in order to differentiate fluorescence within the upper mesh from vascular fluorescence outside the tumor onplant area. Known tumor vessels were selected with the magic wand tool, thus selecting vessels within the mesh area with similar pixilation. The selected vessels were refined until all the tumor vascularization was specifically selected. The area of pixelization divided by the total area of the onplant upper mesh was calculated to determine the percent NV.

**Automated quantification of tumor-induced angiogenesis.** Computer-facilitated quantification of tumor vascularization in each image required identification of foreground red blood vessels within the tumor boundary followed by computing the ratio of pixels deemed to be blood vessels and total pixels within the tumor region. This process was performed using a custom built applet written in R [39] and R-Shiny [40]. Foreground blood vessels (high intensity red pixels) imaged from the $DiIC_{18}(3)$ staining were separated from background (low intensity red pixels) using a threshold which maximizes inter-class variability between the background and foreground groups and minimizes intra-class variance using Otsu's method, a well-established method in image processing for separating background from foreground pixels [41]. Otsu's threshold was computed on the entire image using the R package EBImage [42]. Pixels within the tumor boundary were considered to be blood vessels if their intensity was over Otsu's threshold allowing the computation of the percent of vascularization. This process relied on the additional R packages shinyFiles [43], DT [44] and xlsx [45]. Code for the automated tumor image quantification software is available at https://github.com/rbotts/TumorQuant.

## Statistical analyses

The above automated quantification provides the percent vascularized value for each onplant by quantifying the number of vascular fluorescent pixels that are above the calculated threshold within the upper mesh area (depicted by the yellow outline) and dividing by the total pixel area of the upper mesh outline. Additionally, for validation of the automated quantification tool, we compared the automated quantification with manual quantification for 100 images. The automated and manual calculations of the percent vascularization were compared for each of these images using linear regression analyses and $R^2$ (Fig 2). Once validated, the automated quantification method was used to compute the percent vascularization for all remaining images. Comparisons of the percent vascularization between experimental groups or drug doses was performed using ANOVA, and post-hoc pairwise comparisons using Tukey's method. All tests were performed at the 5% significance level but p-values between groups are indicated as appropriate.

## Compusyn analysis of synergy

To identify synergy, the freely available Compusyn software produced by Ting-Chau Chou was used. This is based on the Median-Effect Equation and the Combination Index Theorem originally described by Chou and Talalay [46]. The doses and resulting effects of bevacizumab alone, temsirolimus alone, and the combination were input and the resulting combination index (CI) values were obtained. In order to arrive at "effect" as required for the program, the average percent tumor vascularization values for the negative control (vehicle) were subtracted from each of the test groups. The relative tumor vascularization for each group was determined by dividing the percent tumor vasculature for each dose by the percent tumor vasculature for the untreated positive control. A result of 90% compared to the untreated control indicated an effect of 10%, or 0.1. A treatment dose / group with only 40% vasculature compared to the positive control was considered a 60% effect (0.6), and so forth.

## Results

### Modification of a standard CAM model for analysis of tumor angiogenesis

In order to screen for combinations of approved angiostatics that provide synergistic activity in preventing tumor-induced neovascularization, a simple and inexpensive model system that

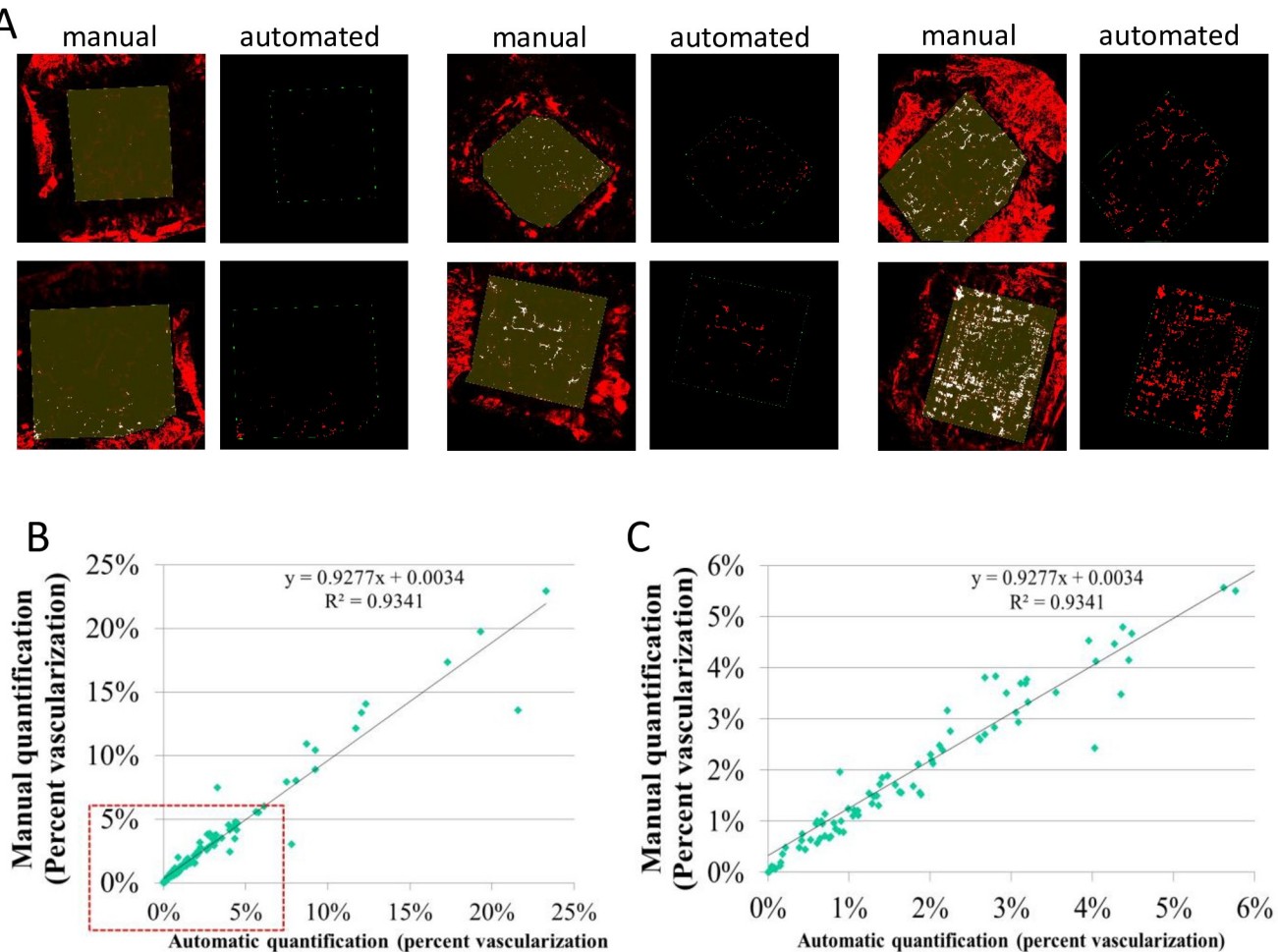

**Fig 2. Comparison of manual quantification vs. automated quantification.** (A) Images from manual quantification are shown (left) with their corresponding images from the automated quantification (right) for 6 samples representative of low, medium, and high levels of NV. Note that in the images from the automated quantifications, vessels outside of the upper mesh area of the onplant are not displayed as a feature of the software. (B—C) Comparison of 100 quantified images using the manual vs. automated computational methods demonstrates strong correlation throughout the range of onplant vascularization (B), as well as a comparison of the percent of vascularization for lower levels of vascularization (below 6%) (C).

allows for testing tumor-induced angiogenesis *in vivo* was required. Various models of tumor angiogenesis have been described [47], but most either require tedious and costly mammalian studies that are not conducive to a large screening project or fail to replicate the complex cellular interactions associated with tumor-induced NV *in vivo*. The chick chorioallantoic membranes (CAM) assay represents an *in vivo* model that: 1) is relatively inexpensive with moderate throughput, 2) utilizes a microenvironment in which tumor angiogenesis can occur naturally, and 3) allows evaluation of either directly or systemically administered antagonists [33, 48].

We developed a modified version of the standard *ex ovo* CAM model described by Quigley and Deryugina [33] (Fig 1). Using two small meshes embedded within the tumor onplants, 3-D tumor onplants with tumor cells and various drug combinations within a collagen 1 matrix were created (Fig 1A' and 1B'). Glioma cells (rat 9L or human U87) were incorporated into the tumor onplants and placed on the CAM vessels of *ex ovo* 10 day chick embryos (Fig 1C and 1D). The tumor cells induced neovascular, right-angle sprouts from the underlying

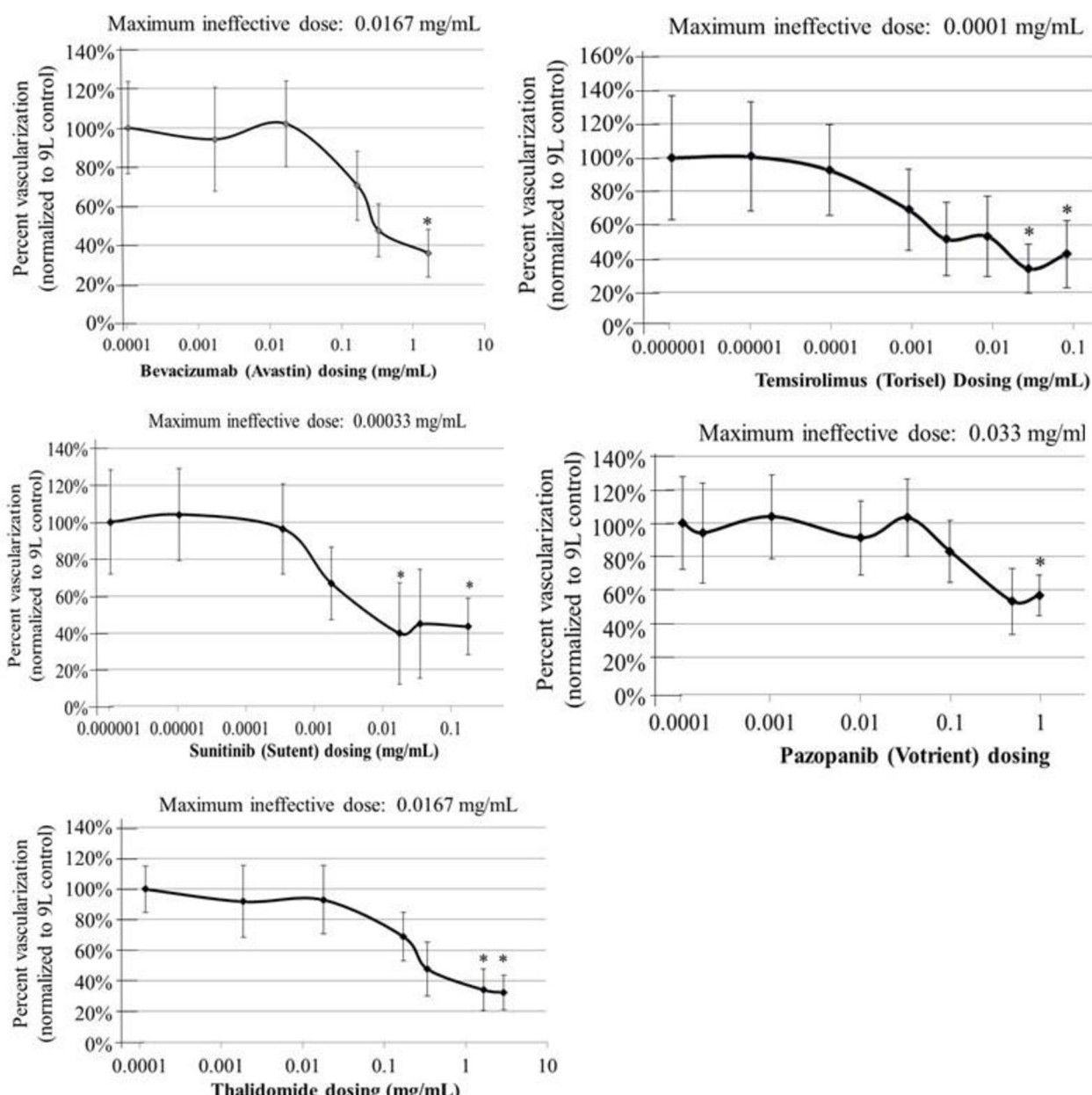

**Fig 3. Monotherapy dosing of angiostatics in the CAM model.** Monotherapies were tested at multiple doses in order to ensure activity based on the inhibition of tumor-induced neovascularization in our model system. Once this functional activity was confirmed, the maximum dose of no effect was identified for each of the various angiostatic monotherapies that demonstrated activity in the CAM model. The indicated maximum ineffective doses were subsequently used to screen for synergistic combinations ensuring that any observed tumor NV inhibiting activity was due to exponential benefits in combination. Asterisks indicate significant reduction in tumor vascularization compared to the untreated controls.

CAM vessels, which subsequently grew into the tumor onplant. Three days later the tumor onplants were excised (Fig 1E) and analyzed for the extent of tumor-induced angiogenesis to quantify the effects of various drugs or combinations of drugs. Confocal microscopy was used to specifically image and quantify the tumor vessels within the z-plane of the upper mesh. By focusing on the upper mesh, tumor angiogenesis was distinguishable from the underlying CAM vessels (Fig 1G). This ensured that only the tumor-derived angiogenic vessels were

quantified; the tumor vasculature had to branch from the CAM vessels at right angles and migrate upward through the lower mesh into the plane of the tumor and upper mesh, thus distinguishing the tumor-initiated vasculature from the underlying, developing CAM vessels (Fig 1B'). During excision of the onplants, CAM vessels around the onplant were included so as not to disrupt the integrity of the tumor-vascular unit within the onplants. These outlying CAM vessels were observed as bright vascular staining around the onplant itself, which was limited to tumor-induced vasculature due to focusing on the upper mesh of the onplant (Fig 1G and 1H). The automated quantification method successfully quantified only the tumor vasculature and was able to quantify the percent area of the tumor onplant that had become vascularized (Fig 1I). Negative control vehicle in collagen 1 alone onplants generally resulted in about 0.5% to 1% vascularization of the onplant area, while tumor induction resulted in vasculature quantified in approximately 4% - 6% of the onplant area demonstrating substantial and significant induction of tumor angiogenesis (Fig 1J).

## Development and analysis of automated methods to quantify neovascularization within the tumor onplants

One of the major limitations to the described confocal quantification of the CAM model was the requirement for manual quantification of the area of NV using Photoshop or ImageJ (Figs 1H and 2A). This method proved to be reliable, but overly time-consuming and user-dependent for efficient screening. There are several good published methods for automating vascular quantification, including some that describe automated quantification of angiogenesis in the CAM model [49, 50]. However, these automated methods quantify the CAM vasculature itself rather than tumor-derived vasculature within tumor onplants placed on top of the CAM. Thus, we developed an automated method for quantifying percent vascularization within the upper mesh area of our tumor onplants based on Otsu's method. By calculating a threshold for each onplant based on background fluorescence intensity of the perfused vasculature and quantifying only within the defined onplant area as indicated by the yellow box, the tumor vasculature was successfully differentiated from background fluorescence and from the outlying CAM vasculature that had been excised with the tumor onplant (Figs 1I and 2A). The number of fluorescent pixels within the yellow-outline delineated upper mesh was determined and divided by the total pixels within the upper mesh area to calculate the percent vascularized area of the onplant (Fig 2A). Comparing the manual quantification and the automated quantification for 100 samples demonstrated a strong correlation between the manual and automated methods with an $R^2$ value of 0.93 and demonstrated accurate quantification for both high and low levels of NV (Fig 2C).

## Identifying the maximum dose of no effect for angiostatic monotherapies in the CAM model

Our goal was to identify effective combinations of currently approved angiostatics, or those in late-stage clinical trial, that confer exponentially improved benefits when combined. Our previous research into combination angiostatics demonstrated that generally the more distinct the mechanism of activity, the more likely two or more angiostatics were to convey synergy [28]. Thus, we sought to identify approved, accessible angiostatics whose mechanisms of action were distinct. Nine different drugs that are currently approved for cancer applications with reported angiostatic activity were obtained and analyzed (Table 1); it should be noted that EGFR antagonists and Torisel® also have other mechanisms of action to prevent cancer growth beyond angiostasis. Bevacizumab, temsirolimus, pazopanib, sutinib, and thalidomide were all found to have angiostatic activity blocking tumor-induced NV in the CAM model

(Fig 3), while cetuximab, sorafenib, erlotinib and the RGD peptide did not demonstrate any significant angiostatic activity in our model system at doses tested (S1 Fig).

We next sought to find the maximum dose of no effect for each angiostatic. Maximum ineffective doses in the CAM model were found to be 16.7 µg/mL for bevacizumab and Thalidomide, 33 µg/mL for pazopanib, 0.33 µg/mL for sunitinib, and 0.1 µg/mL for temsirolimus (Fig 3). Some of the observed doses of no effect are likely to be higher than what might be expected for human clinical use due to lower affinity for interacting with either the chick vasculature or to factors released by the rat 9L tumors. Indeed, bevacizumab has been shown to have 5-fold lower affinity for rat VEGF [51] and Pazopanib has demonstrated slightly lower activity in rat corneal models [52], but both are commonly used in rat models of angiogenesis. Sunitinib and temsirolimus have also been shown to successfully inhibit tumor angiogenesis in rat models of glioma and colorectal cancers [53, 54] although we could not find concrete information on relative affinities for rat versus human for either drug. However, we were not interested in dosing information for the sake of informing clinical doses as such extrapolation would be difficult, but rather for use in subsequent screening to identify synergistic combinations. By combining the drugs at doses with no monotherapeutic effect on tumor angiogenesis, we were subsequently able to determine combinations with synergistic activity since any significantly strong activity in combination is likely to represent benefits beyond those attributed to additive benefits if neither demonstrated any effect alone.

## Bevacizumab and temsirolimus demonstrate synergistic activity in combination

Multiple experiments were performed to screen combinations of drugs using each drug at the maximum ineffective dose to identify combinations with synergistic activity. In each experiment, vehicle control onplants (PBS), untreated tumor onplants, and monotherapy treatment groups were always included. Thus, each separate experiment included at least five groups in order to adequately assess synergy in combination. Most combinations did not result in observed synergy with regards to inhibition of tumor-induced NV in initial tests (S2 Fig). The one combination that resulted in effective inhibition of tumor angiogenesis was bevacizumab and temsirolimus. This combination resulted in reduction of tumor-induced NV similar to the levels observed in control onplants lacking tumor cells (Fig 4). This experimental combination was replicated in a total of six different experiments, with each experiment including over 30 replicates per group. Inclusion of the 9L tumor cells in the onplant resulted in a five-fold increase in NV compared to the non-tumor controls (p-value <0.001). Neither monotherapy alone was able to significantly reduce the tumor-induced NV at these intentionally low, ineffective doses, while the combination together resulted in NV levels below those in the vehicle (no tumor) controls. The vehicle alone control onplants generally induced a small amount of NV, which may be attributed to either natural vascular growth or a small level of stimulation of angiogenesis by the collagen onplant itself. Indeed, collagen 1 has been demonstrated to contribute to angiogenesis [55]. However, the significant, induction of angiogenesis in the 9L tumor control vs. the empty onplant control group demonstrates substantial tumor-specific induction of angiogenesis in our model (Fig 1). The inhibition of tumor-induced NV observed in the combination group was highly significant compared to the tumor control group and both monotherapy control groups across all experiments (p-value <0.01). In addition the CI value for synergy was very low (2.42E-4) (Fig 4). CI values from 0–1 indicate synergy according to Chou's widely used method of quantifying synergy with values closer to zero indicating higher evidence of synergy [46]. Taken together, these results strongly demonstrate a

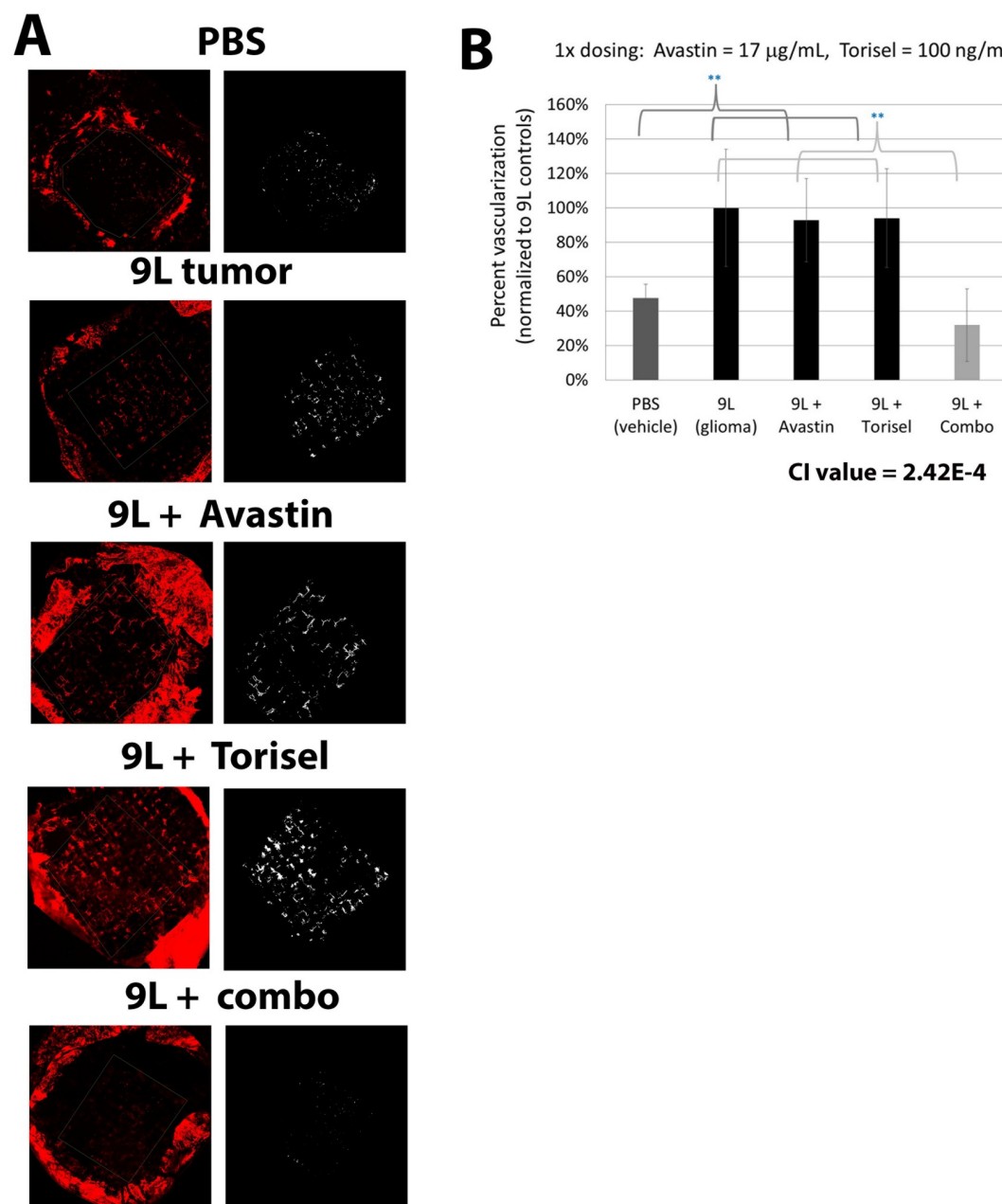

**Fig 4. Combining bevacizumab (Avastin®) and temsirolimus (Torisel®) results in significant activity blocking tumor-induced NV.** (A) Representative images of the tumor onplants from the different treatment groups are shown with the original image on the left and the image showing the results of quantification on the right, followed by quantification of tumor NV (B). As expected, Avastin® and Torisel® did not demonstrate significant reduction in tumor NV individually using the maximum doses of no effect found in the earlier dosing studies (Fig 3). However, significant reduction in tumor-induced angiogenesis was observed when combining Avastin® and Torisel® at these same doses. The resulting angiogenesis levels are comparable to the no-tumor negative control levels. X-axis groups indicate what was included in the collagen onplant. Data include results from six separate experiments with over thirty individual onplants per group per each experiment. Error bars represent standard deviation. ** indicates p values < 0.01 between the indicated groups. The calculated CI value indicates a synergistic effect when combination compared to each monotherapy.

synergistic effect at this dose and indicate that the VEGF inhibitor, bevacizumab (Avastin®) and the mTOR inhibitor, temsirolimus (Torisel®), are exponentially effective at blocking tumor vascularization in combination.

### Bevacizumab and temsirolimus maintain synergistic angiostatic activity when doses are reduced 10-fold

We next tested the combination at doses further reduced 10-fold below those already determined to be ineffective as monotherapies in order to determine the extent of angiostatic synergy and the potential for maintaining efficacy against tumor-induced NV at substantially lower doses. The ability to effectively utilize lower doses of each drug when used in combination could be very important for minimizing side effects during clinical use. Even at these logarithmically reduced doses, the combination of bevacizumab and temsirolimus demonstrated highly effective inhibition of tumor angiogenesis with quantified onplant vascularization nearly identical to onplants devoid of tumor cells (Fig 5). Again, the extent of tumor-induced NV compared to no tumor controls, and the reduction in tumor-induced NV by the combination therapy were highly significant with p-values <0.001. Monotherapies did not inhibit tumor vascularization to any degree and the combination proved to be even more synergistic at this lower dose with a CI value of 2.42E-5. Thus, this combination of bevacizumab and temsirolimus is highly synergistic and 100% effective at blocking 9L glioma-induced angiogenesis in the CAM model, even at concentrations well below doses whereby the individual treatments demonstrate zero activity.

Lastly, we tested the synergistic effectiveness of the combination of bevacizumab (Avastin®) and temsirolimus (Torisel®) in preventing angiogenesis induced by a human glioblastoma tumor line (U87). Similar to the effects observed using 9L glioblastoma cells, bevacizumab and temsirolimus were found to block tumor angiogenesis in a highly effective manner when used in combination, again even at doses ten-fold below their maximum ineffective dose (Fig 5C). In each experiment (Figs 4 and 5), there was a significant increase in tumor NV when the onplants included tumor cells as compared to the PBS control onplants (p-value < 0.001). Each of the monotherapies were statistically similar to the non-treated tumor onplants, while treatment with the combination of bevacizumab and temsirolimus resulted in significant reduction of tumor NV (p-value < 0.001). This demonstrates the effectiveness of this combination in a second glioblasoma cell line, in this case blocking angiogenesis induced by human tumor cells.

## Discussion

We have described a modified CAM model, including a novel computational approach to quantifying the tumor-induced vasculature within the tumor onplants. This was then used to screen and identify approved angiostatics that convey synergistic activity blocking glioma-induced NV when used in combination. It is our hope that these synergistic combinations will overcome some of the clinical limitations that angiostatic monotherapies have had as a primary treatment modality for cancer. By focusing on approved angiostatics, we found a combination that can be more readily tried in clinical settings compared to the use of angiostatics that are not yet approved as monotherapies and, due to a lack of individual benefit over current treatment modalities, are unlikely to be brought to the clinic in the near future [28]. GBM is one of the most aggressive and deadly forms of cancer with a median life expectancy following diagnosis under two years. Fewer than 5% of patients survive for five years post-diagnosis [56]. Thus, new treatment options are particularly needed in aggressive cancers such as these. In this study, we provide evidence that two clinically approved angiostatics, bevacizumab and temsirolimus, synergistically inhibit tumor NV induced by rat or human glioma cells in our *ex ovo* model, even when using doses at 10-fold below the maximum dose of no effect. Clinically, this could translate to reduced tumor metastasis, slowed tumor progression, and improved patient outcome. We tested this using rat and human glioma cell models but our findings may

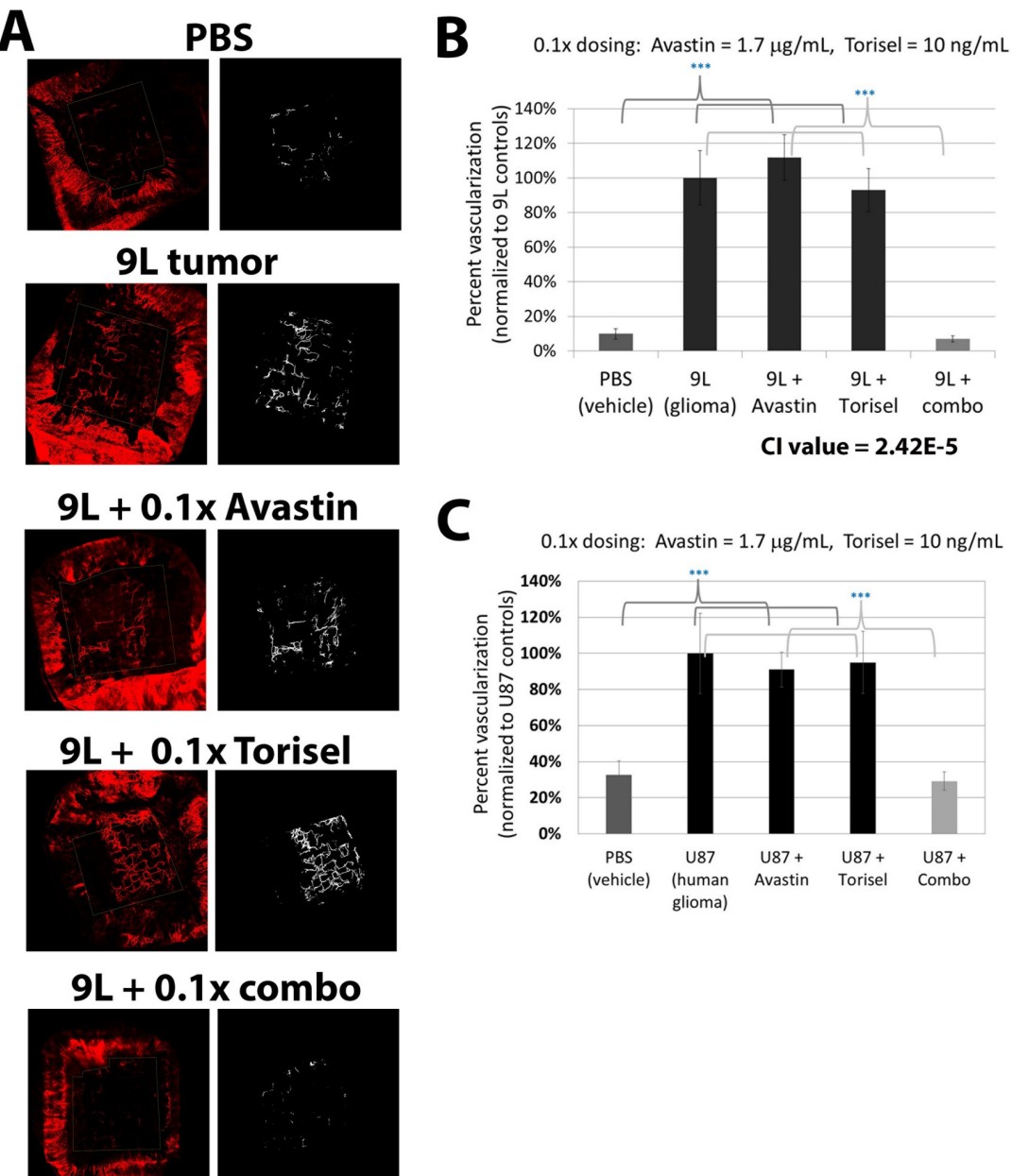

**Fig 5. Synergistic angiostatic activity of bevacizumab and temsirolimus is observed at 10-fold lower doses.** To further demonstrate synergistic activity of combining bevacizumab (Avastin®) and temsirolimus (Torisel®) when blocking tumor-induced NV, each were combined at concentrations ten-fold below their maximum ineffective doses. Individual treatment with Avastin® or Torisel® had no effect on tumor NV, as expected, but in combination, tumor NV was completely blocked. (A) Representative images of the tumor onplants from the different treatment groups are shown with the original image on the left and the image showing the results of quantification on the right (B) Quantification of tumor NV demonstrated strong, synergistic angiostatic activity against tumor-derived NV (CI value = 2.42E-5). (C) Similar results were observed using the human glioblastoma cell line U87. Error bars represent SEM. X-axis groups indicate what was included in the collagen onplant.

be broadly applicable to multiple cancers where NV is critical to tumor progression. While further experiments are required to confirm the combination of bevacizumab and temsirolimus in larger, *in vivo* tumor models, our data demonstrate that their use in combination may help improve the clinical outcome of angiostatic treatments.

One of the limitations to angiostatic therapy has been the ability of tumors to compensate for single-factor inhibition. Previous studies demonstrated that this compensation can be overcome using combinations of distinct angiostatic therapies [28]. However, most combinations are not synergistic. In order to advise doctors of approved angiostatic drugs that effectively block tumor NV in combination, multiple combinations needed to be tested. Thus, combinations of approved angiostatics must be screened, requiring a balance between time / cost and the ability of the particular screening model to adequately represent tumor-induced NV [28]. Here we have presented a novel modification of the CAM assay, used it to identify activity levels of approved angiostatic monotherapies against glioma-induced NV, and tested combinations of those angiostatics to determine which confer synergistic activity. This information will inform further studies that will allow clinicians to combine angiostatic therapies to more effectively treat GBM and potentially other cancers.

Previous reports have demonstrated that the binding of bevacizumb to rodent VEGF is 5-fold weaker to rat VEGF than human VEGF [51]. This may explain the slightly higher dose of bevacizumab required to inhibit angiogenesis induced by 9L (rat) cancer cells in the CAM model compared with other angiostatics. However, many articles have successfully used rodent models of angiogenesis for pre-clinical analysis of bevacizumab [57, 58]. In addition to bevacizumab and temsirolimus, thalidomide, pazopanib and sunitinib also demonstrated significant reduction in tumor-induced NV. While cetuximab, sorafenib, erlotinib hydrochloride, and the RGB peptide did not demonstrate angiostatic activity in our model, there are several potential explanations. We used the generic version of sorafenib and erlotinib hydrochloride, available from Sigma-Aldrich, and thus the results may not represent activities of the clinical formulations in this pre-clinical model. However, it seems more likely that these drugs are not effective in the CAM model system against the rat version of their particular targets, at least not within the range of doses that we tested. It is also possible that the EGFR inhibitors are not effective as angiostatics in this model. There is published evidence of EGFR activity in promoting tumor angiogenesis [59, 60], which is why we included the inhibitors cetuximab and erlotinib hydrochloride in our screen of angiostatics, but EGFR also has many other activities involved in cancer progression including tumor cell growth, proliferation, and survival [61]. Thus, these molecules may be effective anti-cancer agents but not effective when combined to specifically prevent the outcome of tumor angiogenesis.

## Combination therapy of bevacizumab and temsirolimus

Bevacizumab is a monoclonal antibody that blocks vascular endothelial growth factor (VEGF) [62]. As one of the major cytokines that initiates endothelial cell proliferation, VEGF has become one of the key targets of angiostatic therapies. Temsirolimus is a potent inhibitor of the mechanistic target of rapamycin (mTOR) pathway, a key regulator of tumor hypoxia and inducer of several pro-angiogenic growth factors, including VEGF [63]. The pathway also regulates tumor cell proliferation. Therefore, temsirolimus acts not only on the vasculature, but on the cancer cells themselves. Even at doses logarithmically below those that demonstrated any activity as monotherapies, the combination of bevacizumab and temsirolimus demonstrated nearly complete inhibition of tumor-induced angiogenesis when applied in combination. Not only does this suggest that these two drugs may be beneficially used in combination to block pathological, glioma-induced angiogenesis as part of a treatment regimen for patients, it also suggests that the doses required to achieve such strong results may be reduced when used in combination, thus limiting potential side effects. This may prove to overcome many of the disappointing clinical effects of angiostatic monotherapies, and may further reduce unwanted side effects and make such treatments more cost effective.

Many clinical studies have tested the efficacy of angiostatics in combination either with other angiostatics or with various chemo- or immune-therapies. This is particularly true as research has begun to focus on how to prevent escape mechanisms involved in promoting tumor growth and progression in the face of various VEGF inhibitors that have been identified and tested [32]. In the clinical setting, temsirolimus has been shown to have toxic effects on patients, often limiting the recommended dose. In a phase II clinical trial utilizing 250mg/ week of temsirolimus for GBM 36% of patients demonstrated disease regression. However, more than half the patients experienced toxicities that included hypertriglyceridemia, hyper- cholesterolemia, stomatitis, thrombocytopenia and neutropenia [64]. Subsequently there were numerous phase II clinical trials in which the dosing of temsirolimus was lowered to 25 mg/ week in combination with bevacizumab (10 mg/kg every two weeks). For example in one such trial for the treatment of recurrent or persistent endometrial carcinoma, even at the lower dos- ing 38.8% of patients were removed from the study due to toxicity. However, for those able to tolerate the treatment 46.9% of those in this study survived progression free for at least six months [65]. Multiple clinical trials to target renal cell carcinomas (TORAVA and INTOR- ACT) using the same dosing (temsirolimus at 25 mg/week and bevacizumab at 10 mg/kg every two weeks) had large dropout rates due to numerous adverse events including proteinuria, diarrhea, rash and hypertension [66, 67]. A similar regimen of bevacizumab and temsirolimus was used to treat GBM in a phase II clinical trial [68], after patients had previously undergone treatment with temozolomide. In this study specifically looking at recurrent GBM, the combi- nation did not show benefit beyond bevacizumab treatment alone. In light of the numerous adverse events of such treatments, the timing and spacing of the angiostatics appears to be crit- ical for minimizing the toxic effects. Interestingly, a phase II trial for the treatment of recurrent rhabdomyosarcoma with chemotherapeutics (vinorelbine and cyclophosphamide) and either bevacizumab (15 mg/kg every three weeks) or temsirolimus (15mg/m$^2$ every week) was well tolerated, perhaps due to the slightly lower doses and spacing of the treatment cycles [69]. Together these studies illustrate that temsirolimus and bevacizumab although effective when tolerated, are accompanied by significant toxicity issues.

Our studies indicate that effective angiostatic activity against tumor-induced NV can be obtained when utilizing much lower doses of the angiostatics, if appropriate concentrations can be obtained at the site of the tumor. In our study, doses were reduced 10-fold from the maximum dose of no effect, which were already low enough to be completely ineffective indi- vidually. This low dose combination was still able to eliminate tumor-induced angiogenesis. While toxicity is always of highest concern, the ability to lower doses of each drug to logarith- mically reduced levels when applied in combination and still maintain stronger activity than each individually, such as that demonstrated in our study, could be a major benefit of combi- natorial therapy by minimizing toxic side effects. Interestingly recent studies in a murine GBM model utilizing an alternative mTOR inhibitor, RES529, which has been shown to cross the blood-brain barrier, in combination with bevacizumab had reduced tumor progression and increased overall survival [70]. The advent of similarly active compounds that may more easily cross the blood-brain barrier could also increase the potential efficacy of bevacizumab and temsirolimus on glioblastoma brain tumors.

GBM is a highly vascularized tumor that relies on the tumor NV for cancer growth and invasion [71]. In theory, effective angiostatic therapy could overcome some of the issues asso- ciated with treatment of glioblastomas: 1) Unlike the tumor cells themselves, the vasculature is not mutating rapidly and thus is less likely to succumb to drug resistance or tumor-specific drug effectiveness, 2) angiostatic therapy has the potential to limit microsatellite growth since all tumor regions require vessels for extensive growth, and 3) therapy that blocks tumor vascu- larization will affect all tumor cells despite their proliferative status. Finally, since non-

pathogenic angiogenesis is highly limited in adults, the devastating side effects often associated with standard chemotherapy should be limited [72], particularly if combination angiostatic therapy facilitates the use of lower dose chemotherapy than would be required individually. Although an angiostatic approach is not likely to completely eradicate the glioblastoma tumor, if effective, it could offer substantial improvements to the current standard of care. Our data suggests that the combination of VEGF inhibitors, such as bevacizumab, and mTOR pathway inhibitors, such as temsirolimus, should continue to be considered and tested as potential treatments or adjuncts for treatment of glioblastoma and other tumors. This was the only combination tested that demonstrated such strong combinatorial effects on both 9L and U87 glioma-induced angiogenesis in our model.

## Novel adaptation to the CAM model for screening combination angiostatics

The chick chorioallantoic membrane (CAM) assay represents a relatively inexpensive, moderate-throughput *in vivo* model, in which the chorioallantoic vessels that normally support gas exchange near the surface of the egg are used to support tumor growth, thus facilitating analysis of the effects of angiogenesis-regulating factors and antagonists [73]. It has been used to grow tumor onplants, quantify tumor angiogenesis, analyze the effects of antagonists or inflammatory cells on tumor-related angiogenesis [33], and to assess metastatic potential of various tumors [34]. Thus, the CAM model represents a cost-effective, *in vivo* compromise between tissue culture models and mammalian models, and allows multiple therapy options to be screened. There are many in vitro model systems that are used to study angiogenesis. Each of these has their individual strengths and weaknesses [47]. In vitro assays that involve endothelial cell migration or even tube formation assays generally fail to replicate the complex nature of angiogenesis, particularly tumor-derived angiogenesis where the tumor and endothelial cell interactions and the microenvironment play a critical role. While useful, matrigel models generally are not considered representative of angiogenesis by the community since neither the structure nor the mechanisms of formation of the NV into the matrigel matrix are physiologically relevant [47, 74]. Three-dimensional co-culture systems have been created that better replicate the interactions of tumor cells, stromal cells, and endothelial cells [47]. Tissue explant assays such as retinal explants and aortic ring assays allow for the analysis of sprouting of actual vessels from tissue explants and therefore contain more of the vascular-relevant non-endothelial cells and reduce the problems associated with repeated passage of cell lines. However, its main limitations relate to a lack of blood flow in the system and the fact that most angiogenesis occurs from branches off of the arterial explant rather than the venous branches representative of most angiogenesis [47]. Finally, the quantification is generally done by counting the number of vessel branches and can be difficult if the vessel number is high and doesn't account for much of the vascular complexity [75]. In contrast, several *in vivo* model systems are also commonly used to study tumor growth and tumor-induced angiogenesis. Transparent window studies have been used successfully to determine effects of angiostatics on tumors in the natural context with all microenvironmental cells and factors involved [76]. Xenograft tumor explant models also allow for the mechanistic studies of tumor angiogenesis, and analysis of the effects of angiostatics [47]. However, these *in vivo* model systems are highly technical, labor-intensive, and expensive [47] and therefore are not conducive to early screening studies. Despite their *in vivo* nature, they still have their own limitations with regards to tumor angiogenesis, including a lack of normal immune cells in xenograft models involving immune-compromised mice. Thus, these models still tend to over-estimate the effects of anti-tumor drugs [77] and are generally better for pre-clinical confirmation or testing of results from simpler and more cost-effective studies.

The chick chorioallantoic membrane (CAM) assay represents a relatively inexpensive, moderate-throughput *in vivo* model, in which the chorioallantoic vessels that normally support gas exchange near the surface of the egg are used to support growth of tumor onplants, thus facilitating analysis of the effects of angiogenesis-regulating factors and antagonists [73]. Like all assays, it has its own limitations, but it has been used to grow tumor onplants, quantify tumor angiogenesis, analyze the effects of antagonists or inflammatory cells on tumor-related angiogenesis [33], and to assess metastatic potential of various tumors [34]. Thus, the CAM model represents a cost-effective, *in vivo* compromise between tissue culture models and mammalian models, and allows multiple therapy options to be screened.

Some of the best studies for screening effective drugs, including drug combinations and angiostatics, come from studies that combine faster screening models such as the CAM with other models of angiogenesis. One such study by Zhou *et al* describes a four-step system for screening angiogenesis inhibitors [78]. In this study, 480 compounds were tested for angiostatic activity and 28 were found to inhibit angiogenesis. This study demonstrates the power of such models for identifying angiostatic compounds from a chemical library. However, this model system did not study the effects on tumor-induced angiogenesis during the early screen, but rather analyzed effects of drugs on angiogenesis within the CAM vessels themselves and the yolk sac membrane model. After narrowing down the compounds through the initial screens, a single compound was tested for the effects on tumor angiogenesis. This study demonstrates the need to use *in vivo* models to confirm the identified hit following initial screening, but did not screen for combinations of angiostatics in the early screen or subsequent confirmation studies. Our study uses the *ex ovo* CAM model system to stimulate angiogenesis into the tumor onplants for the sake of screening combinations of angiostatics with synergistic activity.

We have described a modified version of the *ex ovo* CAM model that results in automated, quantitative measures of tumor NV. Historically, tumor NV has been quantified in the CAM model by serial cross-sections or by visually counting the percentage of wells that contain an angiogenic vessel within a mesh incorporated into the tumor onplant [33]. These methods are subjective in that they rely extensively on the ability to visualize tiny, un-stained capillaries, and they can also result in inaccuracies since single tiny capillaries are given the same weight as a complex of large, tumor vessels. In order to overcome these difficulties and generate higher throughput and more objective measurements of angiogenesis quantification, we have described a novel method of automated quantification of perfused, fluorescently-labeled tumor angiogenesis using the CAM model. Many automated quantification methods have been described to quantify angiogenesis in various model systems, including the CAM model [49, 50], but these describe methods of quantifying vascularization in the CAM vessels themselves rather than the tumor-induced vasculature. Our model system has the distinct advantage of using natural vessels in their *in vivo* context by taking advantage of the *ex ovo* tumor onplant system, while still focusing solely on tumor-induced NV by focusing on vessels that are induced to grow within the tumor onplant, thus facilitating analysis of angiostatic effects specifically on the tumor vessels. Using this modified, automated method of quantification, we have been able to analyze thousands of different samples across many experiments during our screen with relative efficiency and accuracy. The computation methods for analysis using R Studio are provided freely (https://github.com/rbotts/TumorQuant.) and should provide a new tool for researchers using the CAM model for various angiogenesis-related applications.

## Supporting information

**S1 Fig. Several angiostatics did not demonstrate angiostatic effects in our model system at doses tested.** This was true for cetuximab (Erbitux®), sorafenib (Nexavar®), and erlotinib

hydrochloride (Tarceva®). It is possible that we would observe effects at much higher doses, but these begin to be well beyond the range associated with clinically-related doses and thus would likely to be less relevant for pre-clinical study.
(TIF)

**S2 Fig. Most combinations did not demonstrate synergistic activity.** We tested each possible double combination of the various approved angiostatics that demonstrated activity in our model system. In every combination, each monotherapy was used at the maximum dose of no effect and synergistic activity was assessed in combination.
(TIF)

## Acknowledgments

The authors would like to thank Point Loma Nazarene University and Research Associates for their ongoing support of research scholarship and the summer undergraduate research program. Their vision and generosity has made quality undergraduate research at PLNU possible. This project was performed at an undergraduate teaching institution with many undergraduates key to building this model, acquiring data over many years, replicating experiments, and analyzing data. This article is dedicated to the memory of Dr. Ryan Botts who was a highly valued colleague and friend, and whose brilliance and efforts were critical to this work.

Note that since this project was performed at an undergraduate teaching institution, there were many undergraduates key to building this model, acquiring data over many years, replicating experiments, and analyzing data (hence the large author list).

## Author Contributions

**Conceptualization:** Michael I. Dorrell, Heidi R. Kast-Woelbern, Ryan T. Botts.

**Data curation:** Michael I. Dorrell, Stephen A. Bravo, Jacob R. Tremblay, Sarah Giles, Jessica F. Wada, MaryAnn Alexander, Eric Garcia, Gabriel Villegas, Jordan A. Silva, Bridget M. Fortin, Connor A. Lowey, Allison L. Hale, Troy L. Kurz, Jack C. Rusing, Dawn M. Goral, Paul Thompson, Alec M. Johnson, Daniel J. Elson, Roujih Tadros, Charisa E. Gillette, Carley Coopwood, Amy L. Rausch, Jeffrey M. Snowbarger.

**Formal analysis:** Michael I. Dorrell, Heidi R. Kast-Woelbern, Ryan T. Botts.

**Funding acquisition:** Michael I. Dorrell, Heidi R. Kast-Woelbern.

**Investigation:** Michael I. Dorrell, Heidi R. Kast-Woelbern, Stephen A. Bravo, Jacob R. Tremblay, Sarah Giles, Jessica F. Wada, MaryAnn Alexander, Eric Garcia, Gabriel Villegas, Jordan A. Silva, Bridget M. Fortin, Connor A. Lowey, Allison L. Hale, Troy L. Kurz, Jack C. Rusing, Dawn M. Goral, Paul Thompson, Alec M. Johnson, Daniel J. Elson, Roujih Tadros, Charisa E. Gillette, Carley Coopwood, Amy L. Rausch, Jeffrey M. Snowbarger.

**Methodology:** Michael I. Dorrell, Heidi R. Kast-Woelbern, Ryan T. Botts, Stephen A. Bravo, Jacob R. Tremblay, Sarah Giles, Jessica F. Wada, MaryAnn Alexander, Eric Garcia, Gabriel Villegas, Caylor B. Booth, Kaitlyn J. Purington, Haylie M. Everett, Erik N. Siles, Michael Wheelock, Jordan A. Silva, Bridget M. Fortin, Connor A. Lowey, Allison L. Hale, Troy L. Kurz, Jack C. Rusing, Dawn M. Goral, Paul Thompson, Alec M. Johnson, Daniel J. Elson, Roujih Tadros, Charisa E. Gillette, Carley Coopwood, Amy L. Rausch, Jeffrey M. Snowbarger.

**Project administration:** Michael I. Dorrell, Heidi R. Kast-Woelbern.

**Resources:** Michael I. Dorrell.

**Software:** Ryan T. Botts, Caylor B. Booth, Kaitlyn J. Purington, Haylie M. Everett, Erik N. Siles, Michael Wheelock.

**Supervision:** Michael I. Dorrell, Heidi R. Kast-Woelbern, Ryan T. Botts.

**Validation:** Michael I. Dorrell.

**Writing – original draft:** Michael I. Dorrell, Heidi R. Kast-Woelbern.

**Writing – review & editing:** Michael I. Dorrell, Heidi R. Kast-Woelbern, Ryan T. Botts.

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
