## [Decision Letter · Decision Letter 0]

11 Mar 2021

PONE-D-21-02412

A Novel Method of Screening Combinations of Angiostatics Identifies Bevacizumab and Temsirolimus as Synergistic Inhibitors of Glioma-Induced Angiogenesis

PLOS ONE

Dr. Dorrell,

Thank you for submitting your manuscript to PLOS ONE. After careful consideration, we feel that it has merit but does not fully meet PLOS ONE’s publication criteria as it currently stands. Therefore, we invite you to submit a revised version of the manuscript that addresses the points raised during the review process.

One of the reviewer felt that no comparative parameters are provided with current methods, and why the method in this paper is better than existing methods. Please provide this information. In addition, details regarding methodology and replication of such studies should be provided. Finally, providing a context to your findings in the light of numerous publications reported in the field with combination anti-angiogenesis drug discovery is necessary. Any additional data sets to bolster the conclusion is recommended, and encouraged. 

We look forward to receiving your revised manuscript.

Kind regards,

Ramani Ramchandran

Academic Editor

PLOS ONE

Journal Requirements:

4. Your abstract cannot contain citations. Please only include citations in the body text of the manuscript, and ensure that they remain in ascending numerical order on first mention.

Reviewers' comments:

Reviewer's Responses to Questions

**Comments to the Author**

1. Is the manuscript technically sound, and do the data support the conclusions?

Reviewer #1: Partly

Reviewer #2: Partly

2. Has the statistical analysis been performed appropriately and rigorously? 

Reviewer #1: Yes

Reviewer #2: I Don't Know

3. Have the authors made all data underlying the findings in their manuscript fully available?

Reviewer #1: Yes

Reviewer #2: Yes

4. Is the manuscript presented in an intelligible fashion and written in standard English?

Reviewer #1: Yes

Reviewer #2: Yes

5. Review Comments to the Author

Reviewer #1: The authors utilized an automated tool for measure neovascularization quantification. Screened of Angiostatics combinations on tumor induced neovascularization in CAM model

The authors sated “In this study, we screened multiple combinations of approved angiostatics to identify synergistic combinations that effectively block glioma-induced tumor vascularization similar to the effects observed in our earlier study”.

Please cite appropriate reference.

There is no effect of individual drugs raising the question of bio accessibility from tumor implants. For example: Avastin alone failed to block tumor induced vascularization at any concentration tested. One may be argued that Avastin (a humanized monoclonal antibody may not work on Rat cells) but even small molecules had not shown any effect except the combination. This may not be a synergistic effect.

More data on other combinations of Angistacis could presented

What extent the combo therapy is potent in terms of drug concentration compared to mono therapy could be discussed

Reviewer #2: Dorrell et al, have studied, “A Novel Method of Screening Combinations of Angiostatics Identifies Bevacizumab and Temsirolimus as Synergistic Inhibitors of Glioma-Induced Angiogenesis” using Automated vascular quantification as well as manual vascular quantification using traditional methods. Overall, the study is well conducted, but authors do not exhibit any novel or new information which will need to be published or reported. Combination of many anti-angiogenic drugs are in clinical trials and have copious amount of pre-clinical studies based on their doses and IC values. Here are some examples:

• Temsirolimus and bevacizumab, or sunitinib, or interferon alfa and bevacizumab for patients with advanced renal cell carcinoma (TORAVA): a randomised phase 2 trial

• Randomized Phase II Trial of Bevacizumab or Temsirolimus in Combination with Chemotherapy for First Relapse Rhabdomyosarcoma: A Report from the Children’s Oncology Group

• Anti-tumor effect of bevacizumab on a xenograft model of feline mammary carcinoma

• Controlling escape from angiogenesis inhibitors PMID: 23001349.

• In addition, automated CAM assay, AngioIQ: A Novel Automated Analysis Approach for Angiogenesis Image Quantification, PMID: 17946107; PMID: 24068303: PMID: 25277148, have been widely explored in various setting. Authors have failed to demonstrate advantage of their novel automated vascular quantification over already existed image quantification. How this ‘automated method’ can be utilized to reproduced for other studies, which are specific guidelines/parameters, what is the range of replicates, and false discovery rate etc? Overall, automated method has not been explored in depth to use as standard validation at this level.

6. PLOS authors have the option to publish the peer review history of their article (what does this mean?). If published, this will include your full peer review and any attached files.

Reviewer #1: No

Reviewer #2: No

---

## [Author Response · Author response to Decision Letter 0]

16 Apr 2021

One of the reviewer felt that no comparative parameters are provided with current methods, and why the method in this paper is better than existing methods. Please provide this information. 

We have added extensive sections to the discussion where we compare our model system to other available systems, including the ones suggested by the reviewers. We also added further information putting our study into context with the ongoing clinical and pre-clinical studies regarding combination angiostatics, and particularly some of the clinical studies involving bevacizumab and temsirolimus, the combination that emerged from our screen. 

In addition, details regarding methodology and replication of such studies should be provided.

We have attempted to clarify some of our methodology as part of our clarification for the reviewer’s questions, and we have highlighted our replication of the studies. In fact, our studies of the synergistic combination of Avastin and Torisel was replicated in over 6 different experiment, each with N’s of greater than 30 onplants (at least 6 different CAMs) for each group (negative control onplant with no tumor cells, tumor alone, tumor + Avastin, tumor + Torisel, and tumor + combo). 

Finally, providing a context to your findings in the light of numerous publications reported in the field with combination anti-angiogenesis drug discovery is necessary.

We thank the reviewers for pointing us towards some great articles and studies that we have added to our discussion. We have tried to discuss the vast work of anti-angiogenesis studies both in pre-clinical and clinical studies, including those combinations that have been tested for efficacy against tumor growth. We have added specific discussion of the reviewers’ recommendations. However, there is only so much of this literature we can discuss in this particular context. Our direct relevance is to glioblastoma research and combination angiostatics and we hope we have achieved this context in our discussion. We are happy to include more specific articles or studies if the reviewers have further key specific suggestions that they believe we are missing. 

 Any additional data sets to bolster the conclusion is recommended, and encouraged. 

We have included all of our data for the tested angiostatics that did not demonstrate activity in our model system (S1) and the tested combinations of angiostatics that did not demonstrate synergy during our screen (S2). These are provided as the supplementary figures with the full data provided in the google shared folders with the rest of the data. 

Reviewer #1: The authors utilized an automated tool for measure neovascularization quantification. Screened of Angiostatics combinations on tumor induced neovascularization in CAM model

The authors sated “In this study, we screened multiple combinations of approved angiostatics to identify synergistic combinations that effectively block glioma-induced tumor vascularization similar to the effects observed in our earlier study”.

Please cite appropriate reference.

Thank you for catching this omission. We have added that citation to the text (line 117).

There is no effect of individual drugs raising the question of bio accessibility from tumor implants. For example: Avastin alone failed to block tumor induced vascularization at any concentration tested. One may be argued that Avastin (a humanized monoclonal antibody may not work on Rat cells) but even small molecules had not shown any effect except the combination. 

We were able to see efficacy of Avastin in our model system. We talk about how the efficacy of Avastin may be lower in our model system compared to a human-tumor model system (Discussion paragraph 2). For all of the monotherapies, we tested various doses of each drug by itself to identify doses with efficacy in our CAM model system. As the reviewer states, this is important to ensure activity of the human drug on our rat-tumor (human U87 tumors were used later), which would be expressing rat versions of VEGF, etc. and the chick vasculature. We found that 5 of the drugs had significant efficacy in our model system (Figure 3). These were the only ones tested in various combinations so that we were sure to be screening with molecules that had demonstrated activity in the assay.

Based on this dosing information, we then backed off the dose to the highest dose of no efficacy. This was done so that we could screen for synergistic activity in combination because efficacy of the combination would suggest synergy between the compounds if neither were effective alone at that dose. That is why the monotherapies in figures 4 and 5 didn’t show efficacy at the doses used and why we were excited when the combination showed such strong effects. Each of the monotherapies did show activity at higher doses individually. 

We have further clarified this in the manuscript: “For each individual drug, a broad range of doses were tested until a dose that significantly affected tumor neovascularization in the CAM model was found (Fig 3), thus ensuring efficacy in the CAM model with rat 9L tumor induction (note that a few drugs did not demonstrate efficacy in our model (Table 1)). After initially obtaining broad dosing information, the dosing range was refined to determine the maximum non-effective dose for each angiostatic in the CAM model (Fig 3). These doses were confirmed in at least two additional experiments.” (Materials and Methods; lines 176 – 181)

In the results section: “Bevacizumab, temsirolimus, pazopanib, sutinib, and thalidomide were all found to have activity in the CAM model (Fig 3), while cetuximab, sorafenib, erlotinib and the RGD peptide did not demonstrate any significant angiostatic activity in our model system at doses tested” (Results, lines 379 – 382)

We also added a line in the figure legend to clarify this: “:Monotherapies were tested at multiple doses in order to ensure activity in our model system. Once activity was confirmed, the maximum dose of no effect was identified for each of the various angiostatics that demonstrated activity in the CAM model. The indicated maximum ineffective doses were subsequently used to screen for synergistic combinations. Asterisks indicate significant reduction in tumor vascularization compared to the untreated controls.” (Figure 3 legend: lines 384 – 390)

This may not be a synergistic effect.

We believe we are seeing synergistic effects based on the strong activity in combination at doses that are ineffective as monotherapies (using monotherapies with demonstrated activity in our assay). We have also performed a standard CI analysis for synergy, which strongly indicates a synergistic effect compared to the monotherapies. 

More data on other combinations of Angistatics could presented 

We have now included additional clinical trials utilizing bevacizumab and temsirolimus. Our initial submission included a clinical trial conducted specifically in GBM as well as endometrial cancer. Herein we now include a phase II (TORAVA) and phase III (INTORACT) clinical trial with this combination therapy in renal cell carcinomas (lines 582-585). We also include a phase II study looking at either bevacizumab or temsirolimus at lower doses which seems to be better tolerated for patients in a recurrent rhabdoymyosarcoma model (lines 591-594). 

We also include our data for the combinations that did not demonstrate synergy as part of our S2 figures. 

What extent the combo therapy is potent in terms of drug concentration compared to mono therapy could be discussed

We have shown using two different (10-fold) concentrations of the drugs at which the monotherapies have no effect while the combination reduced tumor-induced neovascularization to control levels, virtually eliminating all tumor-induced NV. We have added several lines to the discussion putting the potential for using lower doses in combination in context, both with clinical efficacy and maximizing drug tolerance in patients. 

We further discuss the important implications of having lower doses still demonstrate potent activity in synergy: “The ability to effectively utilize lower doses of each drug when used in combination could be very important for minimizing side effects during clinical use. Even at doses10-fold below those that were already ineffective as monotherapies, the combination of bevacizumab and temsirolimus demonstrated highly effective inhibition of tumor angiogenesis with quantified onplant vascularization nearly identical to onplants devoid of tumor cells” (lines 465 – 469).

Reviewer #2: Dorrell et al, have studied, “A Novel Method of Screening Combinations of Angiostatics Identifies Bevacizumab and Temsirolimus as Synergistic Inhibitors of Glioma-Induced Angiogenesis” using Automated vascular quantification as well as manual vascular quantification using traditional methods. Overall, the study is well conducted, but authors do not exhibit any novel or new information which will need to be published or reported. 

Thank you for the kind words, but we respectfully disagree that our study does not add to the current field. We undertook this study back in 2009, at a time when our current results would have turned out to be very novel and there was a great need for this type of study. However, we quickly determined that to screen these approved angiostatics, we would need to identify and adapt a model system, thus the modifications of the CAM model, contact and obtain the various drugs from the pharmaceutical manufacturers (where possible) and perform all the tests on individual therapies, followed by screening. Working with undergraduates at a small institution, this have obviously taken longer than desired, but part of that is the amount of work we’ve done to ensure the quality of our study, with many different replications to ensure reproducibility. We understand that during this time a lot of the novelty of this study was lost and that the reasoning behind the length of time for our study should not be a consideration for publication. We do believe that our study still adds to the current work with combination angiostatics and at the very least complements other published studies. We have tried to expound upon this in our revisions to the manuscript, in large part clarifying and comparing based on the reviewers’ insightful suggestions. For example, our novel modification of the CAM model system allows for automated quantification of tumor vasculature within the tumor onplants rather than the underlying CAM vessels. To our knowledge, this is unique in the various models for assessing and quantifying angiogenesis. While it would have been nice for a more novel combination to have emerged from our screen than the Avastin and Torisel combination which has become prominent in the literature recently, the fact that this is the one combination that demonstrated synergistic effects lends confidence to this particular combination and suggests that it further warrants continued investigation at the pre-clinical and clinical levels. Finally, our findings that the doses can be reduced substantially with continued high levels of activity in combination, speaks to the possibility of this combination being used successfully at doses whereby the side effects of Torisel, something that is currently an issue, can be mitigated. We thank the reviewers for their insightful comments and suggestions and for the opportunity to improve our manuscript.

Combination of many anti-angiogenic drugs are in clinical trials and have copious amount of pre-clinical studies based on their doses and IC values. Here are some examples:

• Temsirolimus and bevacizumab, or sunitinib, or interferon alfa and bevacizumab for patients with advanced renal cell carcinoma (TORAVA): a randomised phase 2 trial

This paper is now referenced in the discussion (lines 582-585). 

• Randomized Phase II Trial of Bevacizumab or Temsirolimus in Combination with Chemotherapy for First Relapse Rhabdomyosarcoma: A Report from the Children’s Oncology Group

This paper is now referenced in the discussion (lines 591-594). 

• Anti-tumor effect of bevacizumab on a xenograft model of feline mammary carcinoma

We had previously seen this article and thank the reviewer for bringing it to our attention. However, despite the article’s strength and interesting results in feline mammary carcinoma, we were not able to recognize the direct relationship between this article and our combination therapy model studying glioblastoma in the CAM model for higher throughput screening as there are obviously a lot of great articles on angiogenesis, cancer studies, and tumor models. 

• Controlling escape from angiogenesis inhibitors PMID: 23001349.

This is an excellent review and we have cited it in the introduction as part of our paragraph on compensatory mechanisms and resistance to monotherapy (particularly the many targeting VEGF) and refer to it in both the introduction (lines 109-111) and the discussion (lines 572 – 575) as we try to put our study into context with the multitude of clinical studies on cancer angiogenesis and combinations. 

• In addition, automated CAM assay, AngioIQ: A Novel Automated Analysis Approach for Angiogenesis Image Quantification, PMID: 17946107; PMID: 24068303: PMID: 25277148, have been widely explored in various setting. Authors have failed to demonstrate advantage of their novel automated vascular quantification over already existed image quantification. How this ‘automated method’ can be utilized to reproduced for other studies, which are specific guidelines/parameters, what is the range of replicates, and false discovery rate etc? Overall, automated method has not been explored in depth to use as standard validation at this level.

We thank the reviewer for pointing us in the direction of these automated approaches. We have discussed the two approaches that describe automated angiogenesis quantification in the CAM model in the results section and the discussion. These articles are powerful methods of quantifying CAM angiogenesis, but only quantify the CAM vessels themselves and are not appropriate for analyzing the tumor-induced vasculature that have grown at right angles into the CAM tumor onplants. We have addressed this in the methods section by stating: “There are several good published methods for automated vascular quantification, including some that describe automated quantification of angiogenesis in the CAM model [47, 48]. However, these automated methods are designed to quantify the CAM vasculature itself rather than tumor-derived vasculature within tumor onplants placed on top of the CAM. (lines 344 – 347).

We have added extensive discussion of various angiostatic models (lines 637-664 and 674-687), including those recommended by the reviewers in our analysis of our novel model system (674-687). Entire reviews are available that discuss strengths and weaknesses of angiogenesis assays so we are unable to directly compare every possible assay. However, we hope that our extensive additions have helped to put our model system, particularly the novel adaptations, into better context. We thank the reviewers for pointing out this weakness in our original version of the article and allowing us to improve the discussion of our model system in context of other available options.

---

## [Decision Letter · Decision Letter 1]

4 May 2021

PONE-D-21-02412R1

A Novel Method of Screening Combinations of Angiostatics Identifies Bevacizumab and Temsirolimus as Synergistic Inhibitors of Glioma-Induced Angiogenesis

PLOS ONE

Dear Dr. Dorrell,

Thank you for submitting your manuscript to PLOS ONE. After careful consideration, we feel that it has merit but does not fully meet PLOS ONE’s publication criteria as it currently stands. Therefore, we invite you to submit a revised version of the manuscript that addresses the points raised during the review process.

The reviewers suggest changes to the write-up. Please modify the write up to incorporate their suggestions.

We look forward to receiving your revised manuscript.

Kind regards,

Ramani Ramchandran

Academic Editor

PLOS ONE

Journal Requirements:

Reviewers' comments:

Reviewer's Responses to Questions

**Comments to the Author**

1. If the authors have adequately addressed your comments raised in a previous round of review and you feel that this manuscript is now acceptable for publication, you may indicate that here to bypass the “Comments to the Author” section, enter your conflict of interest statement in the “Confidential to Editor” section, and submit your "Accept" recommendation.

Reviewer #1: All comments have been addressed

Reviewer #2: All comments have been addressed

2. Is the manuscript technically sound, and do the data support the conclusions?

Reviewer #1: Partly

Reviewer #2: Yes

3. Has the statistical analysis been performed appropriately and rigorously? 

Reviewer #1: Yes

Reviewer #2: Yes

4. Have the authors made all data underlying the findings in their manuscript fully available?

Reviewer #1: Yes

Reviewer #2: Yes

5. Is the manuscript presented in an intelligible fashion and written in standard English?

Reviewer #1: Yes

Reviewer #2: No

6. Review Comments to the Author

Reviewer #1: A minor comment:

A repeated high dose of bevacizumab administration could induce antibodies against to bevacizumab in Rat. So it may not be efficient at later time points in the study. the authors could discuss this issue as well.

Reviewer #2: The readability of manuscript has improved tremendously than previous version. However, there are still some issues describing their results/ sub-headings which makes hard to interpret results clearly. It leads to read whole paragraph/section, and then self-assume its context to interpret results. Here are some suggestions.

1. Author should change sub-title of results with specific words in context to their result description. It is hard to interpret with generalized/vague statement for examples

• Maintained activity in combination at lower doses (It is not clear for ‘what activity’, ‘which drugs’)

• Dosing of individual monotherapies (what kind/types of monotherapies)

• Monotherapy dosing in the CAM model (add name of monotherapy if it is reference to single therapy, is CAM model refer to Tumor angiogenesis here?)

• Synergistic activity is observed even at 10-fold lower doses (Please clarify in context ‘Synergistic activity’ which types or class of inhibitors)

• Therapy preparations (what types of therapy? Mono or in combinations)

• Quantification of Synergy (please define clearly’ Synergy’ which kinds of inhibitor refer here)

2. According to their recent response from Authors, ‘For example, our novel modification of the CAM model system allows for automated quantification of tumor vasculature within the tumor onplants rather than the underlying CAM vessels. To our knowledge, this is unique in the various models for assessing and quantifying angiogenesis. Author should clarify about the title reference whether it used in context to normal Angiogenesis or tumor angiogenesis,. It is not clear in the title ‘Automated quantification of angiogenesis’.

7. PLOS authors have the option to publish the peer review history of their article (what does this mean?). If published, this will include your full peer review and any attached files.

Reviewer #1: No

Reviewer #2: No

---

## [Author Response · Author response to Decision Letter 1]

5 May 2021

Reviewer #1: A minor comment:

A repeated high dose of bevacizumab administration could induce antibodies against to bevacizumab in Rat. So it may not be efficient at later time points in the study. the authors could discuss this issue as well.

While this is a good thought, we are not using a rat model to test bevacizumab and only included the efficacy in rat model references to demonstrate that the antibody-based drug was effective against rat glioma VEGF, such as that which would be produced by the 9L tumor cells. 

Reviewer #2: The readability of manuscript has improved tremendously than previous version. However, there are still some issues describing their results/ sub-headings which makes hard to interpret results clearly. It leads to read whole paragraph/section, and then self-assume its context to interpret results. Here are some suggestions.

1. Author should change sub-title of results with specific words in context to their result description. It is hard to interpret with generalized/vague statement for examples

• Maintained activity in combination at lower doses (It is not clear for ‘what activity’, ‘which drugs’)

Line 467 has been modified to “Bevacizumab and temsirolimus maintain synergistic activity when doses are reduced 10-fold”

• Dosing of individual monotherapies (what kind/types of monotherapies)

Line 372 has been modified to “Identifying the maximum dose of no effect for individual 

angiostatic monotherapies” 

• Monotherapy dosing in the CAM model (add name of monotherapy if it is reference to single therapy, is CAM model refer to Tumor angiogenesis here?)

Line 387 has been modified to “Monotherapy dosing of angiostatics in the CAM model”

• Synergistic activity is observed even at 10-fold lower doses (Please clarify in context ‘Synergistic activity’ which types or class of inhibitors)

Line 485 has been modified to “Synergistic activity of bevacizumab and temsirolimus is observed at 10-fold lower doses.” 

• Therapy preparations (what types of therapy? Mono or in combinations) 

Line 174 has been modified to “Preparation of angiostatics for screening”

• Quantification of Synergy (please define clearly’ Synergy’ which kinds of inhibitor refer here)

Line 274 has been modified to “Compusyn analysis of synergy” 

In addition, we altered a few other subheadings to be more specific and represent the main points of each subsection further in line with the reviewer’s suggestions. These are mainly altered in the methods and results sections and can be viewed through the tracked changes.

We also went through the manuscript and attempted to specify “activity” anywhere that was used by adding appropriate reference to “angiostatic activity” or “activity blocking tumor-induced NV.” These minor changes are evident throughout the manuscript.

2. According to their recent response from Authors, ‘For example, our novel modification of the CAM model system allows for automated quantification of tumor vasculature within the tumor onplants rather than the underlying CAM vessels. To our knowledge, this is unique in the various models for assessing and quantifying angiogenesis. Author should clarify about the title reference whether it used in context to normal Angiogenesis or tumor angiogenesis,. It is not clear in the title ‘Automated quantification of angiogenesis’.

Line 341-342 has been modified to “Development and analysis of automated methods to 

quantify neovascularization within the tumor onplants”

---

## [Editor Report · Decision Letter 2]

12 May 2021

A Novel Method of Screening Combinations of Angiostatics Identifies Bevacizumab and Temsirolimus as Synergistic Inhibitors of Glioma-Induced Angiogenesis

PONE-D-21-02412R2

Dr. Dorrell,

We’re pleased to inform you that your manuscript has been judged scientifically suitable for publication and will be formally accepted for publication once it meets all outstanding technical requirements.

Kind regards,

Ramani Ramchandran

Academic Editor

PLOS ONE
---

## [Editor Report · Acceptance letter]

21 May 2021

PONE-D-21-02412R2 

A Novel Method of Screening Combinations of Angiostatics Identifies Bevacizumab and Temsirolimus as Synergistic Inhibitors of Glioma-Induced Angiogenesis 

Dear Dr. Dorrell:

I'm pleased to inform you that your manuscript has been deemed suitable for publication in PLOS ONE. Congratulations! Your manuscript is now with our production department. 

Kind regards, 

on behalf of

Dr. Ramani Ramchandran 

Academic Editor

PLOS ONE